# GPR64, Screened from Ewing Sarcoma Cells, Is a Potential Target for Antibody-Based Therapy for Various Sarcomas

**DOI:** 10.3390/cancers14030814

**Published:** 2022-02-05

**Authors:** Koichi Nakamura, Kunihiro Asanuma, Takayuki Okamoto, Keisuke Yoshida, Yumi Matsuyama, Kouji Kita, Tomohito Hagi, Tomoki Nakamura, Akihiro Sudo

**Affiliations:** 1Department of Orthopedic Surgery, Graduate School of Medicine, Mie University, Tsu 514-8507, Japan; k-nakamura@med.mie-u.ac.jp (K.N.); k-yoshida@clin.medic.mie-u.ac.jp (K.Y.); m-yumi@clin.medic.mie-u.ac.jp (Y.M.); kkita125@gmail.com (K.K.); hagifana@clin.medic.mie-u.ac.jp (T.H.); tomoki66@clin.medic.mie-u.ac.jp (T.N.); a-sudou@clin.medic.mie-u.ac.jp (A.S.); 2Department of Pharmacology, Faculty of Medicine, Shimane University, Izumo 693-8501, Japan; okamoto@med.shimane-u.ac.jp

**Keywords:** Ewing sarcoma, GPR64, antibody-based therapeutics, epididymis, sarcomas

## Abstract

**Simple Summary:**

New strategies for immunotherapy have led to an increased interest in tumor-specific antigens on the cell surface in the field of oncology. Identifying markers in sarcomas is difficult because their tumor mutation burden is less than that of carcinomas. We assumed that a target protein may be acceptable as a therapeutic target, even if it is only expressed in the epididymis along with the tumor, because the epididymis has special barriers, known as the blood–epididymis barrier (BEB). We identified GPR64 as a therapeutic target for Ewing sarcoma via next-generation RNA-sequencing. GPR64 is located on the apical membranes of efferent ductules and separated from antibodies by the BEB. This study revealed, for the first time, that anti-GPR64 antibodies accumulate in various sarcomas and avoid targeting GPR64 in the epididymis in vivo. Furthermore, GPR64 is widely expressed in various sarcomas and is, therefore, a potential antibody-based therapeutic target for sarcomas.

**Abstract:**

Ewing sarcoma is an aggressive and the second most common bone tumor in adolescent and young adult patients. The 5-year survival rate is 60–70% for localized disease but 30% for patients with metastases. Here, we aimed to identify a therapeutic target for Ewing sarcoma and evaluate antibody-based therapeutic agents using in vitro and in vivo models. We identified G protein-coupled receptor 64 (GPR64) as a therapeutic target for Ewing sarcoma via next-generation RNA-sequencing. *GPR64v205* mRNA was expressed in HTB166, A673, MG63, 143B, HS-Sy II, and HT1080 cell lines as well as in Ewing sarcoma, undifferentiated pleomorphic sarcoma, leiomyosarcoma, dedifferentiated liposarcoma, and synovial sarcoma tissues. GPR64 expression was observed in 62.5% of sarcoma cases and was overexpressed in 33.9% cases. GPR64-specific monoclonal antibodies were tested as near-infrared probes for in vivo imaging using subcutaneous tumor mouse xenografts. Fluorescence intensity was stronger for the AF700-labeled anti-GPR64 antibody than that for the AF700-labeled isotype control antibody. GPR64 was detected in engrafted tumors of A673, 143B, HT1080, and the epididymis but not in other resected tissues. The anti-GPR64 antibody showed excellent binding to GPR64-positive tumors but not to healthy tissues. This antibody has potential for drug delivery in the antibody-based treatment of sarcomas.

## 1. Introduction

Ewing sarcoma is an aggressive bone tumor and the second most common bone tumor in children and adolescents [1]. The current standard treatment is a combination of surgery, chemotherapy, and radiotherapy. Significant progress has been made in surgery, chemotherapy, and radiotherapy during the last five decades, leading to a survival rate of approximately 70% in patients with localized disease [2,3]. However, conventional chemotherapy is ineffective in a quarter of patients with localized tumors and three-quarters of patients with metastases. The 5-year survival rate of patients with metastases is no more than 30% [2]. Ewing sarcomas are relatively chemotherapy- and radiation-sensitive. Delivering the anticancer drug or radioactive substance selectively to the tumor can substantially improve treatment outcomes.

Targeted cancer treatment strategies depend on the identification of specific target proteins that enable discrimination between normal and malignant cells [4]. Cell surface proteins are excellent targets for antibody-based therapeutics because of their accessibility [4]. Several tumor cell-killing strategies include antibody-dependent cell-mediated cytotoxicity, complement-dependent cytotoxicity, specific delivery of a cytotoxic payload to tumor cells using antibody-drug conjugates, and antibody-isotope conjugates [5]. T cell chimeric antigen receptor technology has introduced a high degree of tumor selectivity into adoptive cell transfer therapies for treating hematologic malignancies [6]. Recently, a new photodynamic therapy using a monoclonal antibody photoabsorber conjugate has been applied to many solid tumors expressing various cell surface target proteins, and its effectiveness has been proven [7,8]. New strategies for immunotherapies have led to an increased interest in tumor-specific antigens on the cell surface in the field of oncology. The ideal antigens are expressed specifically on tumors to minimize damage to healthy tissues. Proteins expressed in healthy organs are not suitable targets because organ damage is highly predicted. However, it is difficult to identify markers in sarcomas because their tumor mutation burden is less than that of carcinomas [9].

Jennifer et al. reported that LINGO1 is a target in Ewing sarcoma because it is expressed in these tumors and in the brain but not in any other somatic tissues [4]. The brain is largely protected from circulating antibodies by the blood–brain barrier, which is composed of tight junctions. Therefore, a target protein may be acceptable as a therapeutic target, even if it is expressed in the brain along with the tumor. Similarly, the testis and epididymis have special barriers, known as the blood–testis barrier and blood–epididymis barrier (BEB), respectively. The restricted expression and accessible extracellular domain of the cell surface proteins are desired properties for antibody-based therapies. Here, we studied the cell surface proteins expressed in Ewing sarcoma, which are also expressed in healthy tissues, such as the brain, testis, and epididymis, but not in other healthy organs. Previous comprehensive transcriptome analysis of Ewing sarcoma revealed several cell surface antigen candidates targeted by antibody-drug conjugates, but their accumulation in tumors and healthy organs has not been tested in vivo [4]. Our study thus aimed to identify a potential antibody-based therapeutic target for Ewing sarcoma and other sarcomas through in vitro and in vivo experiments.

## 2. Materials and Methods

### 2.1. Cell Culture

Human Ewing sarcoma (HTB166 and A673), human osteosarcoma (143B, MG63, HuO9, HuO9-M112, HuO9-M132, HOS, Saos-2), human synovial sarcoma (HS-SY Ⅱ), human malignant peripheral nerve sheath tumor (HS-PSS, HS-Sch-2), human undifferentiated pleomorphic sarcoma (MFH-ino), and human fibrosarcoma (HT1080) cell lines were used in this study. HT1080, 143B, HOS, and MG63 cells were cultured in modified Eagle’s medium (Gibco, Carlsbad, CA, USA) containing 10% fetal bovine serum (FBS). Saos-2 cells were cultured in McCoy’s 5A (modified) medium (Gibco) containing 15% FBS. HuO9, HuO9-M112, and HuO9-M132 cells were cultured in RPMI-1640 medium (Gibco) containing 10% FBS. HTB166 cells were cultured in RPMI-1640 medium (Gibco) containing 15% FBS. A673, HS-SY Ⅱ, HS-PSS, and HS-Sch-2 cells were cultured in Dulbecco’s modified Eagle medium (DMEM; Gibco) containing 10% FBS and L-glutamine. MFH-ino cells were cultured in DMEM/F12 (Gibco) containing 10% FBS. Cells were maintained as attached monolayers and incubated in a humidified atmosphere with 5% CO_2_ at 37 °C.

### 2.2. RNA Isolation, Library Construction, and High-throughput Sequencing

We used Ewing sarcoma cells (HTB166) as representative sarcoma samples and white blood cells from a healthy volunteer as control samples. Total RNA was extracted using an RNeasy^®^ Mini Kit (Qiagen, Hombrechtikon, Switzerland), and rRNA was deleted using the RiboMinus Eukaryote Kit (Thermo Fisher Scientific, Waltham, MA, USA). Whole-transcriptome sequencing was performed using the Ion Torrent Next-Generation Sequencing System (Thermo Fisher Scientific).

### 2.3. Screening of Promising Drug Targets

We referred to open public resources (Ensemble, The Human Protein Atlas, and UniProtKB) and obtained information about the target candidates, including RNA and protein expression levels in healthy tissues, protein structure, and localization in the cell.

We set the selection criteria for target candidates as follows: (1) possessing a transmembrane domain; (2) mainly expressed in tumors and barely in healthy tissues except for the brain, testis, and epididymis; and (3) possessing a unique epitope in the extracellular domain.

### 2.4. RNA Extraction and cDNA Synthesis

The study was approved (No. 1310) by the Ethics Committee of Mie University Hospital. Written informed consent was obtained from all patients. The design and procedures of the study were carried out in accordance with the principles of the Declaration of Helsinki.

We investigated 14 cell lines and surgically resected tumors from the first cohort of 32 patients. Surgically resected tumors were histologically classified as follows: Ewing sarcoma (*n* = 7), dedifferentiated liposarcoma (*n* = 7), synovial sarcoma (*n* = 7), undifferentiated pleomorphic sarcoma (*n* = 5), leiomyosarcoma (*n* = 4), and adipose tissue (*n* = 2). Total RNA was isolated from 1 × 10^7^ cells or 10–15 mg of fresh-frozen tissues using the RNeasy Mini Kit (Qiagen, Hombrechtikon, Switzerland). DNA contamination was eliminated by treatment with DNase Ι. First-strand cDNA was synthesized using the First-Strand cDNA Synthesis Kit (Roche, Basel, Switzerland).

### 2.5. PCR Analysis

PCR was performed using Tks Gflex DNA Polymerase (Takara, Shiga, Japan) in a thermal cycler (Applied Biosystems, Foster City, CA, USA). Primer (Eurofins Genomics K.K., Luxembourg, France) sequences were as follows: human GPR64v205, forward 5′ CCCCCTCCTCCAATGAGGTT 3′ and reverse 5′ GCCTCTGCTGTAGCACACAT 3′. The forward primer was designed for a specific region of the GPR64v205 mRNA (the boundary between exons 5 and 6) (Figure 1). The reverse primer was designed for exon 13 of the same mRNA. The amplification parameters were as follows: 35 cycles of denaturation at 98 °C for 15 s, annealing at 60 °C for 15 s, and extension at 68 °C for 30 s. PCR products were separated via electrophoresis on a 2.0% agarose gel.

### 2.6. Direct Sequencing of the GPR64 Splicing Variant Cassette (Exons 5 to 13)

To ensure that hybridization occurred between the PCR primers and exons 5 to 13 of *GPR64v205* cDNA, PCR products were visualized using agarose gel electrophoresis. DNA fragments (355 bp; predicted band size) were extracted from the agarose gel and purified using a PCR Clean-up Gel Extraction Kit (MACHEREY-NAGEL, North Rhine-Westphalia, Germany). Purified PCR products were treated with Big Dye 3.1. and directly sequenced using a Genetic Analyzer AB3130xl (Applied Biosystems).

### 2.7. Immunohistochemistry of Surgical Specimens

We investigated the second cohort of 112 tumor tissues surgically resected between 1998 and 2019 at Mie University Hospital. Tumor tissue sections were subjected to immunohistochemical analyses. Tumor blocks were formalin-fixed, paraffin-embedded, and sliced into 4 μm-thick sections. This was followed by subsequent incubation in methanol containing 1% H_2_O_2_ for 30 min at 24 °C to eliminate endogenous peroxidase activity. Antigen retrieval was performed by heating the sections for 10 min in citrate buffer (pH 6.0) using an autoclave sterilizer. The slides were incubated with a primary antibody specific for GPR64 (#AF7977, polyclonal sheep anti-GPR64, 3 μg/mL, R&D System Inc., Minneapolis, MN, USA) overnight at 24 °C. After washing, the slides were incubated with horseradish peroxidase-coupled secondary antibody (#A130-101P, polyclonal donkey anti-IgG-heavy and light chain antibody, Bethyl Laboratories, Inc., Montgomery, TX, USA) for 30 min at 24 °C. Following three additional washes, all specimens were stained with 3,3′-diaminobenzidine (DAB; Dojindo Laboratories, Kumamoto, Japan). Finally, the sections were rinsed in distilled water and counterstained with Mayer’s hematoxylin, according to the manufacturer’s instructions.

### 2.8. Quantification of Immunohistochemical Staining for Surgical Specimens

Expression was assessed semi-quantitatively using two parameters: Staining intensity and the percentage of stained tumor cells, as suggested previously [10,11]. The immunohistochemical staining intensity was rated as follows: 0, negative; 1, weak; 2, intense. Density was rated as follows: 0 point, 0%; 1 point, 1–50%; 2 points, 51–75%; and 3 points, 76–100% of positive tumor cells (Figure 2). The eventual scores of each specimen were calculated by adding the intensity and density scores. Expression levels of GPR64 were finally determined as negative (score = 0), low (score ≤ 2), or high (score ≥ 3). If the two assessments did not match, the sample was re-evaluated and classified based on the evaluations given most frequently by the experts. All the whole western blot figures can be found in the Appendix A.

### 2.9. Western Blot Analysis

Lysates extracted from A673, HT1080, and 143B cells were lysed using radioimmunoprecipitation (RIPA) buffer (Millipore-Upstate, Temecula, CA, USA) supplemented with a protease inhibitor cocktail. They were diluted in SDS sample buffer (0.5 M Tris-HCl, pH 6.8, 10% sodium dodecyl sulfate, 30% glycerol, 9.3% dithiothreitol, and 0.00012% bromophenol blue) at a ratio of 1:5, boiled for 5 min, and stored at −80 °C. The samples were separated via 10% SDS-polyacrylamide gel electrophoresis and transferred onto polyvinylidene difluoride (PVDF) membranes (Millipore Corporation, Bedford, MA, USA). After blocking the membrane with TBS-T (20 mM Tris-HCl [pH 7.6] and 1% Tween-20) containing 5% non-fat dried milk for 1 h at 24 °C, it was incubated with antibodies against GPR64 (#AF7977, 2 μg/mL of 1% BSA). β-actin was used as a loading control.

### 2.10. Immunocytochemistry

Immunocytochemistry was performed to investigate the binding specificities of GPR64 and GPR64-specific antibodies. HTB166, HT1080, 143B, and A673 cells were collected at 1 × 10^6^ cells and diluted in 0.1 mL of PBS. Subsequently, the cells were incubated with an anti-GPR64 antibody (#AF7977, 0.25 μg diluted in 1.2 μL of 1% BSA). HTB166, HT1080, 143B, and A673 cells resuspended in 0.1 mL of PBS and incubated with 1.2 μL of 1% BSA as a negative control. The cells were incubated for 30 min at 37 °C (5% CO_2_). After incubation, the reaction medium was centrifuged (5 min, 1500 rpm) and aspirated. The cells were then suspended in PBS and centrifuged (5 min, 1500 rpm) three times. Next, the cells were incubated with an Alexa Fluor 488 (AF488)-labeled anti-sheep secondary antibody (1:400) (Invitrogen, Waltham, MA, USA) for 30 min at 37 °C (5% CO_2_) in the dark. After incubation, the reaction medium was centrifuged (5 min, 1500 rpm) and aspirated. The cells were suspended in PBS, centrifuged (5 min, 1500 rpm) three times, and imaged using a fluorescence microscope (Olympus, Tokyo, Japan).

### 2.11. Xenograft Model

Five-week-old male BALB/c nu/nu mice (Charles River Laboratories International, Wilmington, MA, USA) were maintained in a humidity- and temperature-controlled laminar flow room. For xenografting, 2 × 10^6^ A673, HT1080, or 143B cells in 0.1 mL of saline were subcutaneously injected into the backs of nude mice, using a 26-gauge needle. Tumors were allowed to grow to 8–12 mm in diameter before fluorescence imaging. All experimental procedures involving mice were approved by the Institutional Committee on Animal Welfare.

### 2.12. In Vivo Near-infrared (NIR) Optical Imaging

To analyze the specific target accumulation of imaging probes, nude mice were anesthetized with a combination of 0.3 mg/kg medetomidine, 4.0 mg/kg midazolam, and 5.0 mg/kg butorphanol. Anesthetics were administered to the mice via intraperitoneal injection. Mice with dorsal tumors (*n* = 6) were randomly divided into experimental and control groups, with three mice per group. Each mouse in the experimental group was injected with 200 µL of PBS containing 20 μg of Alexa Fluor 700-labeled (AF700) (Ex/Em = 675–700 nm/723 nm) anti-GPR64 antibody (clone# 864238, 20 μg/mouse) (#MAB79771) (R&D System Inc.) via the tail vein, whereas each mouse in the control group received the same volume of AF700-labeled IgG1 isotype control (clone# MOPC-21, 20 μg/mouse) (SONY, Tokyo, Japan). Surface fluorescence images of the dorsal side of mice were acquired using the IVIS^®^ Lumina III (Perkin Elmer, Inc., Waltham, MA, USA). A 660 nm excitation filter and a 710 nm emission filter were selected. All fluorescence images were presented using the manufacturer’s software (Living Image^®^ software, version 4.4). Mouse whole-body images were acquired using a charge-coupled device camera at 1, 12, 24, and 48 h time points after tail-vein injection of the AF700-conjugated antibody.

### 2.13. Ex Vivo NIR Optical Imaging and Immunohistochemistry

After in vivo fluorescence imaging, the mice were sacrificed immediately using an overdose of a combination of medetomidine, midazolam, and butorphanol as described above. Tumors and organs (the brain, heart, liver, spleen, lung, kidney, epididymis, trachea, blood, quadriceps femoris muscle, skin, salivary gland, pancreas, stomach, colon, and small intestine) were harvested and washed with saline solution. Ex vivo fluorescence images of the tumors and tissues were obtained using IVIS Spectrum^®^ under the same parameters stated above. Exactly 48 h after injecting the AF700-conjugated anti-GPR64 antibody or IgG1 isotype control antibody, the fluorescence intensities from each isolated tissue were measured using a fiber-optic device, and the signal intensities from the tumors were compared with those from other tissues. Fluorescence signal intensities were compared among tissues treated with AF700-conjugated anti-GPR64 antibody and corresponding tissues treated with AF700-conjugated IgG1 isotype control antibody. To investigate the expression of GPR64 in tissue samples, paraffin-embedded samples resected from mice injected with IgG1 isotype control antibody were prepared for immunohistochemistry. GPR64 expression was evaluated via immunohistochemistry using a primary antibody against GPR64 (#AF7977) (polyclonal sheep anti-GPR64, 3 μg/mL, R&D System Inc., MN, USA) with horseradish peroxidase-coupled secondary antibody (#A130-101P, Bethyl Laboratories, Inc.) as described above.

### 2.14. Fluorescence Image Analysis

On the fluorescence images, regions of interest were drawn carefully along the margins of tumors and tissues in vivo and ex vivo. The mean fluorescence intensities (average radiant efficiency, (p/s/cm^2^/sr)/(µW/cm^2^)) were generated automatically using Living Image^®^ 4.7.3 software and recorded.

### 2.15. Statistical Analysis

All statistical analyses were performed using the EZR graphical user interface (Saitama Medical Center, Jichi Medical University, Saitama, Japan) for R (The R Foundation for Statistical Computing, Vienna, Austria), which is a modified version of R commander designed to add the statistical functions frequently used in biostatistics [12].

Data are expressed as the mean ± standard error. A t-test was used to compare the means of two groups. Differences were considered statistically significant at *p* < 0.05.

## 3. Results

### 3.1. Cell Surface Protein Data Reveal Therapeutic Targets for Ewing Sarcoma

The total number of RNA transcripts in the Ewing sarcoma and white blood cells were 230,756. We set the criteria such that the expression level of Ewing sarcoma-related proteins had a value more than 3 and that of white blood cell proteins had a value less than 1. After sorting, 4577 transcripts were obtained. Among these transcripts, we reviewed the localization of the translated proteins in tissues, protein sequences, and extracellular sequences. Finally, we concluded that *GPR64v205* and downstream GPR64 isoform 2 were promising molecular targets because of their high expression at the mRNA and protein levels and limited expression in healthy tissues (2 February 2022, https://www.proteinatlas.org/ENSG00000173698-ADGRG2/tissue) (Table 1). These were specifically expressed in the epididymis [13]. GPR64 had a seven-transmembrane subunit with a unique structure of the large extracellular domain (Figure 3). Alternative splicing generated 11 types of splicing variants, which were translated into nine types of splice isoforms (according to the UniProtKB and Ensembl genome browser). The GPR64 isoform 2 differed from the canonical sequence (isoform 1) because of the lack of amino acids 65–67. Next, we examined the expression of GPR64 at the RNA and protein levels using cell lines and tumor tissues.

### 3.2. Primer Design of GPR64v205 and Direct Sequencing of the PCR Product

To detect *GPR64v205* mRNA, specific PCR primers were designed: a forward primer that bound to the boundary between exons 5 to 6 and a reverse primer that bound to exon 13 because GTACTCCAG on exon 5 was deleted in *GPR64v205* (Figure 1). cDNA derived from HTB166 cells was used as the PCR template. Gel electrophoresis revealed a DNA band of approximately 355 bp in size. The sequence of amplified PCR products matched the reference sequence of *GPR64v205* (350/355: 98.5%) and the specific sequence of *GPR64v205* with a nucleotide deletion in the cranial sequence. We confirmed that the PCR primer pairs could specifically detect *GPR64v205*.

### 3.3. GPR64v205 mRNA Is Expressed in Sarcomas

PCR analyses revealed that *GPR64v205* was expressed in HTB166, A673, Saos-2, MG63, 143B, HOS, HS-Sy II, and HT1080 cells but was weakly expressed in healthy adipose tissues (Figure 4). *GPR64v205* expression was observed in five (71.4%) of seven Ewing sarcoma tissues, three (75%) of four leiomyosarcomas, three (60%) of five undifferentiated pleomorphic sarcomas, five (71.4%) of seven synovial sarcomas, and six (85.7%) of seven dedifferentiated liposarcomas.

### 3.4. GPR64 Protein Is Expressed in Sarcoma Tissues

Immunohistochemistry revealed GPR64 expression in 62.5% (70/112) of sarcoma cases, with high expression in 33.9% (38/112) cases. High GPR64 expression was observed in 90.9% (10/11) of Ewing sarcoma, 36.3% (4/11) of chondrosarcoma, 30.7% (4/13) of osteosarcoma, 80.0% (12/15) of chordoma, 50% (2/4) of fibrosarcoma (FS), 16.6% (2/12) of myxofibrosarcoma (MFS), 11.1% (1/9) of leiomyosarcoma (LMS), 15.4% (2/13) of dedifferentiated liposarcoma, 100% (1/1) of pleomorphic liposarcoma, 0% (0/3) of malignant peripheral nerve sheath tumor (MPNST), 0% (0/3) of myxoid liposarcoma (MLPS), 0% (0/13) of undifferentiated pleomorphic sarcoma (UPS), and 0% (0/4) of synovial sarcoma tissues (Table 2).

### 3.5. Sarcoma Cell Lines for In Vivo Study Express GPR64 Protein

Lysates of A673, HT1080, and 143B cells were subjected to immunoblotting using antibodies against GPR64 and β-actin. Immunoblotting with the anti-GPR64 antibody showed a band for full-length GPR64 at 150 kDa. Western blotting results showed that full-length GPR64 expression was higher in lysates prepared from A673 than in those from 143B and HT1080 cells (Figure 5).

### 3.6. Binding Capacity of the Anti-GPR64 Antibody under Non-Denaturing Conditions

To investigate the binding between the antibody and GPR64 on the cell, HTB166, A673, HT1080, and 143B cells under non-denaturing conditions were stained with sheep anti-GPR64 antibody (#AF7977) and AF488-labeled anti-sheep secondary antibody.

Fluorescence microscopy revealed that the anti-GPR64 antibody was bound to a protein on the HTB166, A673, HT1080, and 143B cells, whereas the AF488-labeled secondary antibody did not show specific binding (Figure 6). We confirmed antigen–antibody reactions between native GPR64 and anti-GPR64 antibodies in vitro.

### 3.7. In Vivo NIR Optical Imaging

A673, HT1080, and 143B cells were used in the in vivo study because PCR and immunostaining data revealed GPR64 expression in these cells and they were successfully engrafted in nude mice. HTB166 cells were not used because they could not be engrafted in nude mice. Whole-body imaging revealed that A673 and HT1080 cells had different fluorescence signal intensities compared with the anti-GPR64 antibody injection group and IgG1 isotype control antibody injection group in vivo (Figure 7A,B). The anti-GPR64 antibody showed strong average radiant efficiency in A673- and HT1080-engrafted tumors at 24 and 48 h time points (*p* = 0.0273 and *p* = 0.0418; *p* = 0.006 and *p* = 0.026, respectively) (Figure 7A,B). Localization of the average radiant efficiency in the tumor area became increasingly evident and intensified from 24 to 48 h in A673- and HT1080-engrafted tumors. In contrast, the average radiant efficiency in 143B-engrafted tumors showed no significant difference between the anti-GPR64 antibody injection group and the IgG1 isotype control injection group at any time point in vivo (Figure 7C).

### 3.8. Ex Vivo NIR Optical Imaging

In the experimental group, the mean fluorescence intensities of A673 tumors were higher than those of the other tissues. The mean fluorescence intensity of A673 tumors was 6.7 times stronger in the experimental group than that in the control group (*p* = 0.0488) (Figure 7A). Low fluorescence intensities were observed in the healthy tissues. The fluorescence intensities of tissues from healthy organs, except for the liver, were not significantly different between the experimental and control groups. The signal intensity of the epididymis was as high as that of the other healthy tissues. In the control group, fluorescence signal intensities around the tumor area were almost equal to those in other tissues.

The mean fluorescence intensity of HT1080 tumors was 2.8 times higher in the experimental group than that in the control groups (*p* = 0.03) (Figure 7B). High fluorescence intensities were observed in the livers of the experimental group mice. However, the mean fluorescence intensities of healthy organs, including the liver, were not significantly different between the experimental and control groups. Ex vivo 143B tumors in the experimental group significantly showed 1.8 times higher fluorescence intensity than the corresponding 143B tumors in the control group (*p* = 0.02) (Figure 7C). High fluorescence intensities were detected in the liver, stomach, small intestine, and skin. However, there was no significant difference between the corresponding healthy organs in the experimental and control groups.

### 3.9. Immunohistochemical Staining of the Resected Tissues

We also immunohistochemically evaluated the expression of GPR64 in resected specimens from the control group to determine whether the signal intensities from the in vivo and ex vivo imaging of the tumors and other tissues reflect the expression levels of GPR64. GPR64 was detected in engrafted tumors of A673, 143B, HT1080, and in the epididymis but not in other resected healthy tissues (Figure 8).

## 4. Discussion

### 4.1. G Protein-Coupled Receptors, GPR64, and Ewing Sarcoma

Drug-targeted G protein-coupled receptors (GPCRs) account for approximately 27% of the global market share of therapeutic drugs [14]. GPCRs are excellent therapeutic targets because of their substantial involvement in human pathophysiology and pharmacological traceability [14]. GPCRs have unique and variable extracellular domains that can be easily targeted using antibodies as diagnostic and therapeutic tools [15]. In general, the terminal regions of the extracellular domain are suitable as antigens owing to their hydrophilicity and structural flexibility.

GPR64 expression was first reported to be upregulated in Ewing sarcoma in 2003 [16]. It is associated with the invasiveness and metastasis of Ewing sarcoma via the regulation of PGF and MMP1 [16]. A previous study showed that GPR64 is a cell surface antigen of Ewing sarcoma in vitro [4]. However, to the best of our knowledge, no studies have examined whether anti-GPR64 antibodies accumulate in GPR64-positive tumors in vivo and determined the expression rate of GPR64 extensively within the range of sarcomas. We report, for the first time, the potential of the anti-GPR64 antibody as an antibody-based therapeutic in sarcomas.

### 4.2. Alternative Splicing

Alternative splicing generates multiple mRNAs and downstream proteins from a single gene through the inclusion or exclusion of specific exons. These mRNAs are called splice variants, and the downstream proteins are called splice isoforms. This process occurs in 95% of all multi-exonic genes, and aberrant splicing allows the production of noncanonical and tumor-specific splice variants in tumor tissues [17]. In cancer, aberrant expression of splice variants is associated with tumorigenesis [18,19]. Alternative splicing of *GPR64* pre-mRNA generates 11 splice variants, resulting in the production of nine types of splice isoforms (Figure 3). Alternative splicing occurs intensively around the N-terminal region, which generates a variety of splice isoforms with slightly different extracellular domains [20]. There are several commercially available anti-GPR64 antibodies, with immunogens usually established from a common sequence between the isoforms. In this study, the expression of *GPR64v205* was upregulated more than that of any other *GPR64* splice variant in Ewing sarcoma. GPR64 isoform 2 was expressed in Ewing sarcoma tissues. It is thus rational to seek epitopes in the N-terminal region of GPR64 isoform 2 because of its structural flexibility, hydrophilicity, and expression level. The anti-GPR64 antibodies used in this study are commercially available. We thus investigated the potential of the anti-GPR64 antibody as a drug delivery tool.

### 4.3. RNA Expression Rate

The expression rate of GPR64 at the RNA level has been reported in the public sarcoma gene expression library as follows: 94.7% (18/19) for Ewing sarcoma, 16.7% (1/6) for fibrosarcoma, 18.2% (4/22) for liposarcoma, 22.2% (6/22) for UPS; not detected for MPNST (0/3), osteosarcoma (0/5), synovial sarcoma (0/14), and leiomyosarcoma (0/15) [16,21]. In this study, *GPR64* cDNA positivity was 71.4% for Ewing sarcoma, 60% for UPS, 75% for leiomyosarcoma, 85.7% for dedifferentiated liposarcoma, and 71.4% for synovial sarcoma. There were some differences between our findings and the public sarcoma gene expression database. This difference may have resulted from the heterogeneity of sarcomas, race, or other unknown reasons.

### 4.4. Protein Expression Rate

High GPR64 expression has already been reported in prostate, kidney, and non-small cell lung cancers as well as in melanoma. In contrast, ovarian cancer, breast cancer, colon cancer, and leukemia show moderate to low expression levels. In this study, high expression of GPR64 was observed in Ewing sarcoma (90.9% [11/11]), chordoma (80.0% [12/15]), osteosarcoma (30.7% [4/13]), chondrosarcoma (36.3% [4/11]), fibrosarcoma (50% [2/4]), myxofibrosarcoma (16.6% [2/12]), dedifferentiated liposarcoma (15.4% [2/13]), and leiomyosarcoma (11.1% [1/9]) (Table 2). High GPR64 expression was observed mainly in bone tumors. This indicated that GPR64 has potential as a therapeutic target in Ewing sarcoma as well as in other bone and soft tissue sarcomas.

### 4.5. Immunocytochemistry

The conformation and glycosylation of cell surface proteins are preserved in vivo. Fluorescence microscopy revealed that the polyclonal anti-GPR64 antibody bound to GPR64 on the cell surface without denaturation, as observed in vitro (Figure 6). This result suggested that the extracellular domain of natural GPR64 has an antibody-binding site. To increase the specificity of the antigen–antibody binding, we used a monoclonal anti-GPR64 antibody (#MAB79771) produced from the same immunogenic peptides as the polyclonal anti-GPR64 antibody (#AF7977) via in vivo experiments.

### 4.6. NIR Imaging

Specific targeting of anti-GPR64 antibody was demonstrated by comparing NIR images of the experimental group with those of the control group. The result was consistent with that of the in vitro study.

The uptake of AF700-labeled anti-GPR64 antibody was strong for A673 tumors, moderate for HT1080 tumors, and weak for 143B tumors in the experimental groups in vivo and ex vivo (Figure 7). Western blotting showed that GPR64 expression was higher in A673 cells than in HT1080 and 143B cells (Figure 5). Immunohistochemistry also revealed that resected tumors showed high expression of GPR64 in A673, low expression in HT1080, and low expression in 143B tumors. The uptake of AF700-labeled anti-GPR64 antibody in tumors was largely dependent on GPR64 expression. This result suggests that anti-GPR64 antibodies accumulate around GPR64-positive tumors via the antigen–antibody interaction. In contrast, high uptake of AF700 was observed in the livers of the experimental and control groups, but GPR64 was not expressed in the liver. We speculate that the high fluorescence signal intensities in the liver are caused by the metabolism of monoclonal antibodies [22]. None of the healthy tissues, except for the liver, showed a significant difference in fluorescence signal intensities between the experimental and control groups ex vivo.

### 4.7. BEB

Many toxic reactions in immunotherapy occur because of the expression of the cognate antigen in healthy tissues. GPR64 is not expressed in essential tissues but is highly expressed in the epididymis [20]. GPR64 regulates fluid reabsorption and is involved in spermatozoon maturation in the epididymis [23]. The influence of chemotherapy or radiotherapy on the genital organ is not negligible because Ewing sarcoma is the second most common malignant bone tumor in adolescent and young adult patients. Efferent ductules of the epididymis possess a unique structure, termed the BEB. The BEB is composed of tight junctions between epithelial cells (Figure 9) [24]. This barrier restricts the transport of toxic or harmful molecules, macromolecules, and antibodies to protect the maturing spermatozoa from the immune system [20,24,25]. GPR64 is located on the apical membranes of efferent ductules and is separated from antibodies by the BEB. In this study, the resected epididymis showed high expression of GPR64 at the apical membranes of efferent ductules. Ex vivo NIR imaging revealed that the fluorescence intensities in the epididymis were not significantly different between the experimental and control groups. They were almost equal in organs other than the liver and tumors in the experimental group. To the best of our knowledge, this study was the first to unravel the unique properties of an anti-GPR64 antibody that avoids targeting GPR64 in the epididymis in vivo.

### 4.8. Limitations

GPR64 is not expressed in human female organs according to public resources, and female mice were not used in this study. The immunogen of the GPR64 antibodies used in this study has a 513-amino acid sequence, which is quite long and, therefore, has low specificity and may not be able to differentiate isoform 2 from different GPR64 isoforms. However, all GPR64 isoforms are expressed on epididymal duct apical membranes and, therefore, pose no problem even if GPR64 isoform 2 cannot be differentiated from other GPR64 isoforms in the application of antibody-based therapeutics [20].

The PCR results indicate the presence of *GPR64* mRNA in A673, HT1080, and 143B cells; however, western blotting showed lower protein expression in HT1080 and 143B than in A673 cells. GPR64 is cleaved by a conserved GPCR proteolytic site (GPS) domain [20,26]. The anti-GPR64 antibodies used in this study could not recognize cleaved GPR64 on the cell surface as stated in the immunogen data provided by the manufacturer. GPR64 may thus be cleaved at the GPS domain on the HT1080 and 143B cells at a certain rate, although the antibodies (#AF7977) cannot detect cleaved GPR64 on the cell via western blotting. The extent to which GPR64 cleavage attenuates the binding efficiency of the antibody to GPR64-positive tumors in this study remains unclear; therefore, further investigation is required.

Although it would have been interesting to explore the antitumor effect of the isotope conjugated anti-GPR64 antibody, our facility prohibits the administration of isotope-labeled drugs to animals in principle; therefore, we did not conduct these experiments.

## 5. Conclusions

This study revealed, for the first time, that antibodies against GPR64 accumulate in bone and soft tissue sarcomas in vivo and do not accumulate in the epididymis in vivo. In addition, we showed that GPR64 is widely expressed in various sarcomas and is therefore, expected to be widely applied in the treatment of sarcomas. We demonstrated that GPR64 shows excellent properties as an immunotherapeutic target. The anti-GPR64 antibody used in this study is commercially available and a promising drug delivery tool for antibody-based therapeutics.

## Figures and Tables

**Figure 1 cancers-14-00814-f001:**
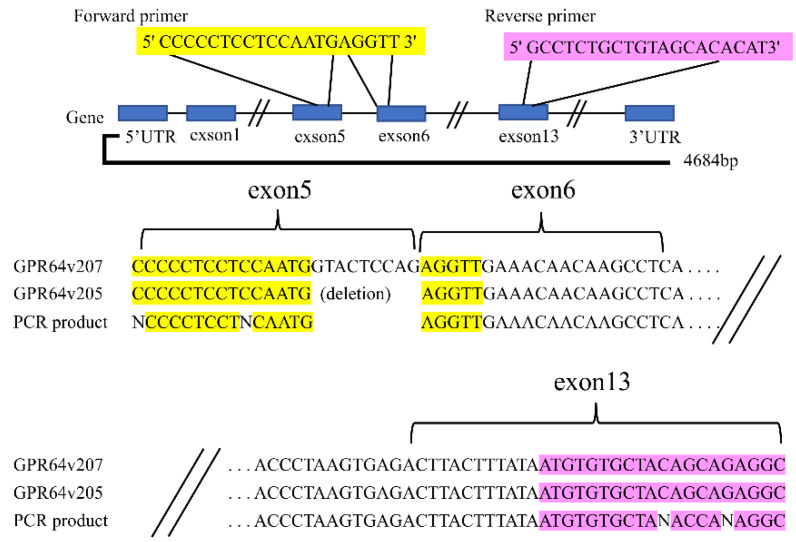
Direct sequencing of PCR products shows consistent results with the *GPR64v205* reference sequence.

**Figure 2 cancers-14-00814-f002:**
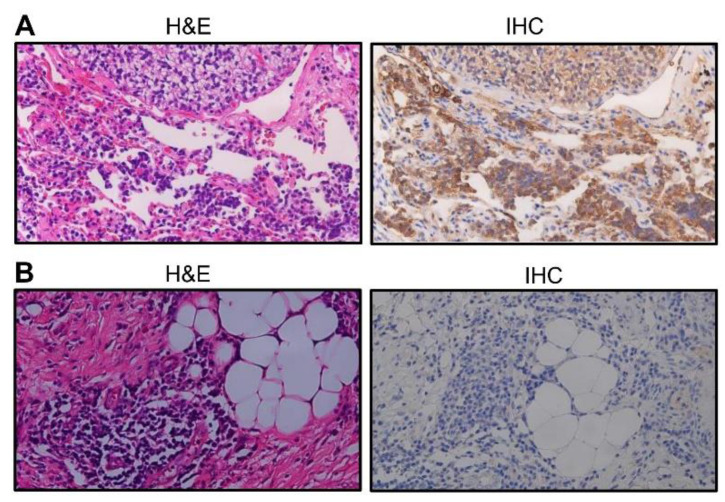
Immunohistochemistry with the anti-GPR64 antibody. (**A**) High expression (Ewing sarcoma tissue samples). Density = 3, Intensity = 2, Total Score = 5. (**B**) Representative data of negative expression (MPNST tissue samples). Density = 0, Intensity = 0, Total Score = 0. IHC, immunohistochemistry; H&E, hematoxylin and eosin.

**Figure 3 cancers-14-00814-f003:**
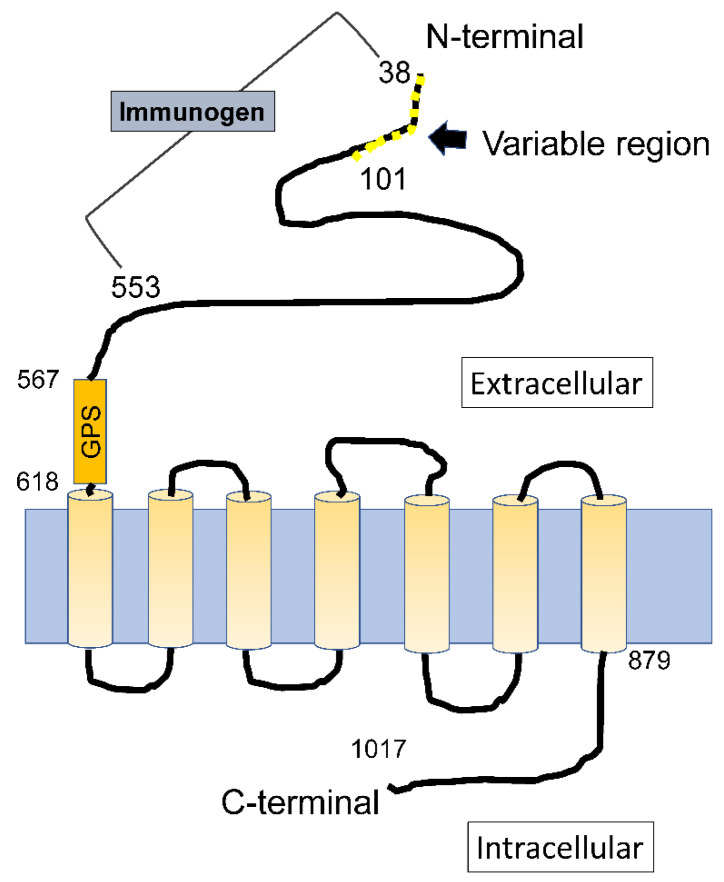
The sequence comprising amino acids 38–101 is called the variable region. Selective splicing occurs intensively, and diversity is observed mainly in this area. Isoform 1, full length; isoform 2, lacking amino acids 65–67; isoform 3, lacking amino acids 51–66; isoform 4, lacking amino acids 88–101; isoform 5, lacking amino acids 52–75; isoform 6, amino acids 52–101 sequence change; isoform 7, amino acids 52–101 sequence change; isoform 9, lacking amino acids 906–956; and isoform 10, lacking amino acids 51–66, 88–101, 474–562. Modified from Kirchhoff et al., 2006.

**Figure 4 cancers-14-00814-f004:**
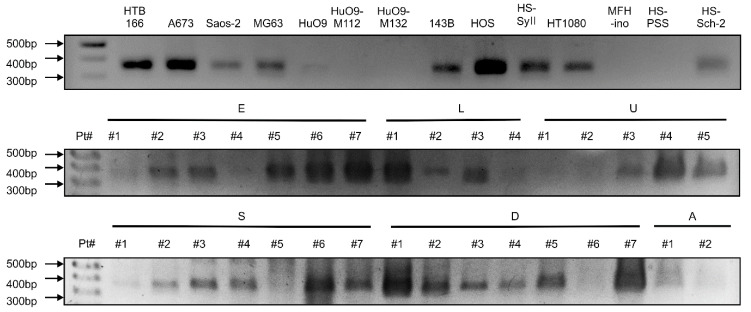
GPR64 is differentially expressed in Ewing sarcoma and other sarcoma tissues and cell lines. E, patient with Ewing sarcoma; L, patient with leiomyosarcoma; U, patient with undifferentiated pleomorphic sarcoma; S, patient with synovial sarcoma; D, patient with dedifferentiated liposarcoma; A, adipose tissues; #, patient number.

**Figure 5 cancers-14-00814-f005:**
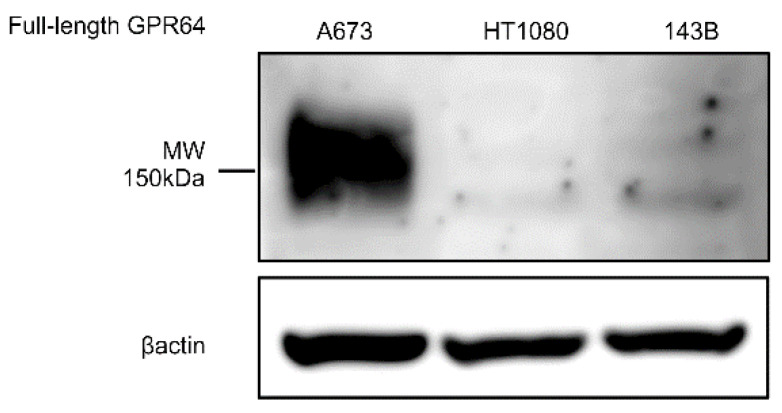
Immunoblotting of tumor lysates extracted from A673, HT1080, and 143B cells. Antibodies against GPR64 and β-actin were used. A specific band was detected for GPR64 at approximately 150 kDa.

**Figure 6 cancers-14-00814-f006:**
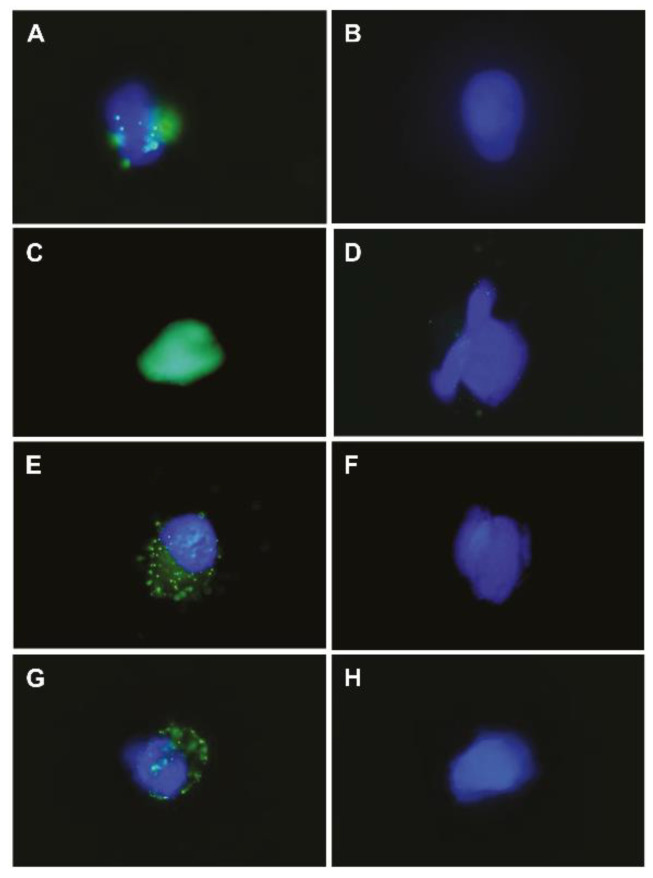
Fluorescence photomicrograph showing binding of the anti-GPR64 antibody to human sarcoma cell lines in vitro. (**A**) HTB166+sheep anti-GPR64 IgG antibody+AF488-labeled anti-sheep antibody. (**B**) HTB166+AF488-labeled anti-sheep antibody. (**C**) HT1080+sheep anti-GPR64 IgG antibody+AF488-labeled anti-sheep antibody. (**D**) HT1080+AF488-labeled anti-sheep antibody. (**E**) 143B+sheep anti-GPR64 IgG antibody+AF488-labeled anti-sheep antibody. (**F**) 143B+AF488-labeled anti-sheep antibody. (**G**) A673+sheep anti-GPR64 IgG antibody+AF488-labeled anti-sheep antibody. (**H**) A673+AF488-labeled anti-sheep antibody.

**Figure 7 cancers-14-00814-f007:**
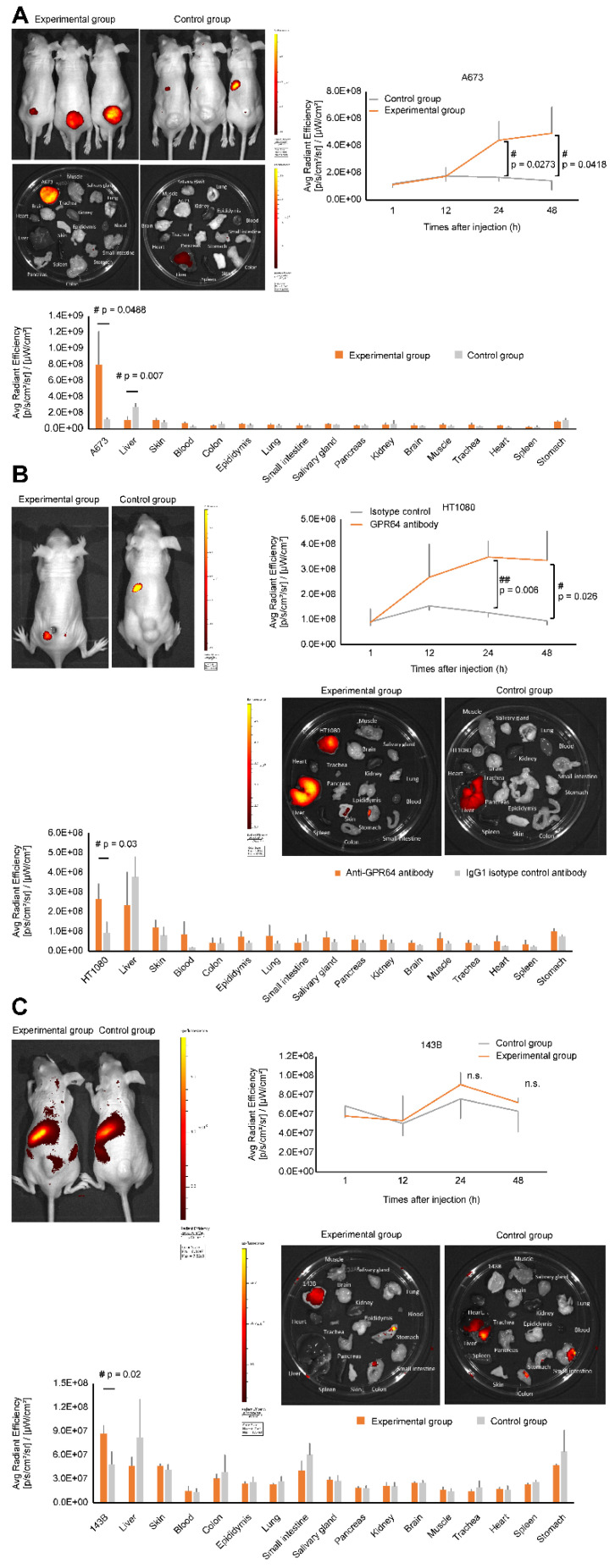
In vivo near infrared (NIR) optical imaging of human sarcoma cell xenograft-bearing mice. Alexa Fluor 700 (AF700)-conjugated GPR64 monoclonal antibody and isotype control antibody were used in this experiment. The average radiant efficiency of engrafted tumors was measured at different time points after tail-vein injecting the antibodies. The experimental and control groups had three mice per group. (**A**) Mice with A673-engrafted tumors 48 h after tail-vein injection (**top left**). Ex vivo NIR fluorescence imaging of the removed A673-engrafted tumor and organs (**middle**). Average radiant efficiency of the A673-engrafted tumors at different time points (**top right**) and the removed organs at 48 h (**bottom**). (**B**) Representative mice with HT1080-engrafted tumors at 48 h after tail-vein injection (**top left**). Ex vivo NIR fluorescence imaging of the removed HT1080-engrafted tumor and organs (**middle**). Average radiant efficiency of HT1080-engrafted tumors at different time points (**top right**) and removed organs at 48 h (**bottom**). (**C**) Representative mice with 143B-engrafted tumors at 48 h after tail-vein injection (**top left**). Ex vivo NIR fluorescence imaging of the removed 143B-engrafted tumor and organs (**middle**). Average radiant efficiency of 143B-engrafted tumors at different time points (**top right**) and removed organs at 48 h (**bottom**).

**Figure 8 cancers-14-00814-f008:**
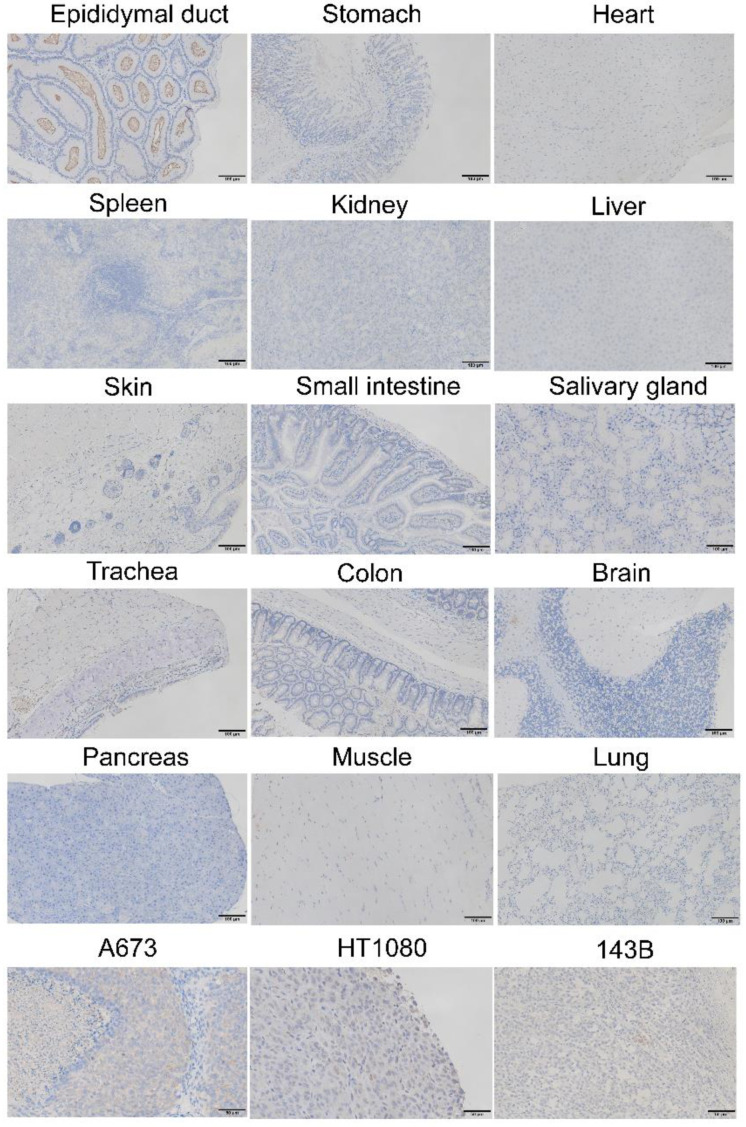
GPR64 expression in tumors and normal organs resected from IgG1 isotype control antibody-injected mice. Tumor (A673, HT1080, 143B): original magnification, ×400. Normal organs: original magnification, ×200.

**Figure 9 cancers-14-00814-f009:**
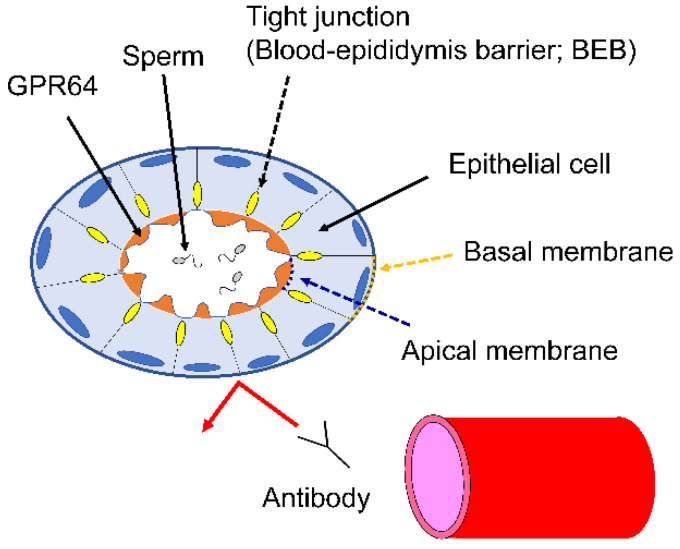
GPR64 on the apical membrane of epithelial cells is protected from the immune system by the blood–epididymis barrier.

**Table 1 cancers-14-00814-t001:** Expression level of *GPR64* splicing variants in HTB166 cells. *GPR64v205* expression is upregulated the most among all *GPR64* splice variants. *GPR64v205* is translated into GPR64 isoform 2.

Splice Variant	HTB166 (FPKM)	WBC (FPKM)	Protein Isoform	Protein Length (aa)
*GPR64v201*	1.15869	0	10	898
*GPR64v202*	1.69 × 10^−8^	1.58 × 10^−5^	10	898
*GPR64v203*	3.4776	0	4	1003
*GPR64v204*	2.26204	9.20 × 10^−82^	7	987
*GPR64v205*	5.29482	0	2	1014
*GPR64v206*	0.0028735	1.32 × 10^−82^	6	995
*GPR64v207*	3.90 × 10^−8^	6.73 × 10^−85^	1	1017
*GPR64v208*	3.50 × 10^−6^	0.0684216	9	966
*GPR64v209*	4.29718	0.000105219	5	993
*GPR64v210*	1.49 × 10^−6^	1.58 × 10^−5^	3	1001
*GPR64v211*	0.419505	0	No protein	

FPKM, fragments per kilobase of exon per million reads mapped; WBC, white blood cell.

**Table 2 cancers-14-00814-t002:** GPR64 isoform 2 expression in sarcomas.

Diagnosis	*n*	Positive Rate (%)	High Expression Rate (%)
Ewing sarcoma	11	100	90.9
Osteosarcoma	13	76.9	30.7
Chondrosarcoma	11	90.9	36.3
PLPS	1	100	100
Myxofibrosarcoma	12	50	16.6
Chordoma	15	93.3	80.0
UPS	13	46.1	0
Synovial sarcoma	4	0	0
DLPS	13	53.8	15.4
Leiomyosarcoma	9	33.3	11.1
Fibrosarcoma	4	50	50
MPNST	3	0	0
MLPS	3	0	0
Total	112	62.5	33.9

PLPS, pleomorphic liposarcoma; UPS, undifferentiated pleomorphic sarcoma; DLPS, dedifferentiated liposarcoma; MPNST, malignant peripheral nerve sheath; MLPS, myxoid liposarcoma.

## Data Availability

The data generated in this study are available upon request from the corresponding author.

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
