# Peer review of "GPR64, Screened from Ewing Sarcoma Cells, Is a Potential Target for Antibody-Based Therapy for Various Sarcomas"

_cancers, 2022, doi:10.3390/cancers14030814_

Round 1

Reviewer 1 Report

This is in my opinion a very nice study and I have only a few minor comments:

  1. The authors should explain what GPR64 is in the Abstract (i.e. a G-protein coupled receptor);
  2. 2.  The end of the intro is a bit unclear (including some grammatical mistakes);
  3. I am not fond of the term "cell surfaceome" for cell surface proteins, but I noticed that there were 3,100 Goggle hits. 
  4. 3.6. The binding... without denaturing". This is not well expressed, why not "under non-denaturing conditions" or "Binding of antibody to native GPR64" or something similar?
  5. Fig 7 and 8: the number of animals/tumors in each group are not given in the legend (one has to consult the M&M for this). In general, legends are quite short and spartan. The readers would benefit from more info.
  6. Fig 8: these are the same tumors as in Fig 7? If so, I would have preferred fusing Fig 7 and 8 into one figure.

Author Response

1. [The authors should explain what GPR64 is in the Abstract (i.e., a G-protein coupled receptor);]

Response: We have added definition of GPR64 to the Abstract.

2. [The end of the intro is a bit unclear (including some grammatical mistakes);

I am not fond of the term "cell surfaceome" for cell surface proteins, but I noticed that there were 3,100 Goggle hits.]

Response: We have changed the phrase “cell surfaceome data” to “cell surface protein data”.

3. [The binding... without denaturing". This is not well expressed, why not "under non-denaturing conditions" or "Binding of antibody to native GPR64" or something similar?]

Response: The suggested key terms have been used throughout the manuscript.

4. [Fig 7 and 8: the number of animals/tumors in each group are not given in the legend (one has to consult the M&M for this). In general, legends are quite short and spartan. The readers would benefit from more info. Fig 8: these are the same tumors as in Fig 7? If so, I would have preferred fusing Fig 7 and 8 into one figure.]

Response: We have corrected the grammatical issues in both the manuscript and accompanying figures. The number of animals/tumors is described in figure 7 legend. I have fused figure 7 and figure 8 into one figure.

Reviewer 2 Report

The present manuscript is well written, titled: “GPR64, screened from Ewing sarcoma cells, is a potential target for antibody-based therapy for various sarcomas”. However, the manuscript lacks connecting data from one result to another. Data needs to be refined in a presentable manner. The manuscript would be strengthened by addressing the points below.

Major comments:

  1. Authors need to show the GPR64 role in the tumor growth, EMT phenotype of cancer cells. And need to show the patient survival analysis of various sarcomas with respect to the GPR64.
  2. Authors need to re-think to conduct the antibody therapy to protect the epithelial cells from the immune system in immunocompromised nude mice.
  3. Authors need to show the effectiveness of anti-GPR64 antibody therapy in sarcomas and the molecular mechanism behind it.

Author Response

1-1. [Authors need to show the GPR64 role in the tumor growth, EMT phenotype of cancer cells.]

Response:

A previous study reported GPR64 maintains an immature phenotype that is less sensitive to TNF-related apoptosis-inducing ligand (TRAIL)-induced apoptosis as it upregulates PGF and MMP1 [1]. However, the role of GPR64 in the tumor growth mechanism is not still elucidated completely.

The study does not focus on determining a target to block molecular function of GPR64. In contrast, the purpose of the study is to discover antibodies against tumor-specific antigens delivered to the tumors specifically in vivo. Ultimately, it is intended for application to antibody-based therapeutics such as antibody-conjugated cytotoxic drugs or isotopes (“armed antibody”).

  1. Richter, G.H.; Fasan, A.; Hauer, K.; Grunewald, T.G.; Berns, C.; Rössler, S.; Naumann, I.; Staege, M.S.; Fulda, S.; Esposito, I. G‐Protein coupled receptor 64 promotes invasiveness and metastasis in Ewing sarcomas through PGF and MMP1. The Journal of pathology. 2013, 230, 70-81.

1-2[And need to show the patient survival analysis of various sarcomas with respect to the GPR64.]

Response: The survival analysis with respect to the GPR64 is interesting but we did not investigate this, as we focus on the potential of anti-GPR64 antibody as a drug delivery tool and the application scope of the treatment of sarcoma. The anti-tumor effect of the antibody itself, is not the main subject of our research.

2. [Authors need to re-think to conduct the antibody therapy to protect the epithelial cells from the immune system in immunocompromised nude mice.]

Response:

Tight junction between epididymal epithelial cells is composed of the blood-epididymis barrier (BEB). The BEB sequesters sperm antigens and restricts entry of immunoglobulins and immune cells, and it protects the sperm from attack by immune system [2].

GPR64 is only expressed in the apical membrane of the epididymis [3]. Therefore, GPR64, as well as the sperm, is protected from the antibody by BEB. This study shows that anti-GPR64 antibody does not cause an accumulation of GPR64 on the luminal side of the epididymal duct in mice. This fact is independent of the presence or absence of immunity in mice.

  1. Mital, P.; Hinton, B.T.; Dufour, J.M. The blood-testis and blood-epididymis barriers are more than just their tight junctions. Biol Reprod. 2011, 84, 851-858.
  2. Kirchhoff, C.; Osterhoff, C.; Samalecos, A. HE6/GPR64 adhesion receptor co-localizes with apical and subapical F-actin scaffold in male excurrent duct epithelia. Reproduction. 2008, 136, 235-246.

3. [Authors need to show the effectiveness of anti-GPR64 antibody therapy in sarcomas and the molecular mechanism behind it.]

Response:  The aim of this study was to investigate the potential of the anti-GPR64 antibody as a drug delivery tool. We did not intend to elucidate the anti-tumor effect of this antibody itself. It would have been interesting to explore the antitumor effect of the isotope conjugated anti-GPR64 antibody; however, our facility prohibits the administration of isotope-labeled antibodies to animals; thus, and we cannot conduct these experiments.

Round 2

Reviewer 2 Report

Authors have addressed all comments, at their best. I am satisfied with their response to my comments. The present manuscript is in acceptable form.